# Is increased mortality by multiple exposures to COVID-19 an overseen factor when aiming for herd immunity?

Kristina Barbara Helle[1☉], Arlinda Sadiku[1☉], Girma Mesfin Zelleke[2,3☉], Toheeb Babatunde Ibrahim[1,2☉], Aliou Bouba[1,2], Henri Christian Tsoungui Obama Junior[1], Vincent Appiah[5], Gideon Akumah Ngwa[1,3], Miranda Ijang Teboh-Ewungkem[4], Kristan Alexander Schneider[1☉]*

**1** Department of Applied Computer- and Biosciences, University of Applied Sciences Mittweida, Mittweida, Germany, **2** African Institute for Mathematical Sciences Cameroon, Limbe, Cameroon, **3** Department of Mathematics, University of Buea, Buea, Cameroon, **4** Department of Mathematics, Lehigh University, Bethlehem, PA, United States of America, **5** West African Centre for Cell Biology of Infectious Pathogens, University of Ghana, Accra, Ghana

☉ These authors contributed equally to this work.
* kristan.schneider@hs-mittweida.de

**Data Availability Statement:** All relevant data are within the paper and its Supporting information files. The implementation of the model can be

## Abstract

### Background

Governments across the globe responded with different strategies to the COVID-19 pandemic. While some countries adopted measures, which have been perceived controversial, others pursued a strategy aiming for herd immunity. The latter is even more controversial and has been called unethical by the WHO Director-General. Inevitably, without proper control measures, viral diversity increases and multiple infectious exposures become common, when the pandemic reaches its maximum. This harbors not only a potential threat overseen by simplified theoretical arguments in support of herd immunity, but also deserves attention when assessing response measures to increasing numbers of infection.

### Methods and findings

We extend the simulation model underlying the pandemic preparedness web interface CovidSim 1.1 (http://covidsim.eu/) to study the hypothetical effect of increased morbidity and mortality due to 'multi-infections', either acquired at by successive infective contacts during the course of one infection or by a single infective contact with a multi-infected individual. The simulations are adjusted to reflect roughly the situation in the USA. We assume a phase of general contact reduction ("lockdown") at the beginning of the epidemic and additional case-isolation measures. We study the hypothetical effects of varying enhancements in morbidity and mortality, different likelihoods of multi-infected individuals to spread multi-infections and different susceptibility to multi-infections in different disease phases. It is demonstrated that multi-infections lead to a slight reduction in the number of infections, as these are more likely to get isolated due to their higher morbidity. However, the latter substantially increases the number of deaths. Furthermore, simulations indicate that a potential

found at GitHub https://github.com/Maths-against-Malaria/MultipleExposuresCOVID19. All data can be reproduced by this code and the parameters specified in the figures and tables in the main text and Supporting information.

**Funding:** K. A. S. could not have put together the research team without the supported by the German Academic Exchange (DAAD; https://www.daad.de/de/; Project-ID 57417782), the Sächsisches Staatsministerium für Wissenschaft, Kultur und Tourismus and Sächsische Aufbaubank – Förderbank (SMWK-SAB; https://www.smwk.sachsen.de/; https://www.sab.sachsen.de/; project "Innovationsvorhaben zur Profilschärfung an Hochschulen für angewandte Wissenschaften", Project-ID 100257255), the Federal Ministry of Education and Research (BMBF) and the DLR (Project-ID 01DQ20002; https://www.bmbf.de/; https://www.dlr.de/). G. N. is supported by the German Academic Exchange (DAAD; https://www.daad.de/de/; Project-ID 57479556) to K.A.S. K.B.H. was supported by the ESF-SMWK-SAB (ESF-SMWK-SAB, https://www.esf.de/portal/DE/Startseite/inhalt.html, https://www.strukturfonds.sachsen.de/europaeischer-sozialfonds-esf.html, https://www.smwk.sachsen.de/, https://www.sab.sachsen.de/; "Nachwuchsforschergruppe - Agile Publika"; Project 100310497) A.S. is supported by a scholarship from the ESF-SMWK-SAB ESF-SMWK-SAB, https://www.esf.de/portal/DE/Startseite/inhalt.html, https://www.strukturfonds.sachsen.de/europaeischer-sozialfonds-esf.html, https://www.smwk.sachsen.de/, https://www.sab.sachsen.de/; "Nachwuchsforschergruppe - Agile Publika"; Project 1562608885160) The funders had no role in study design, data collection and analysis, decision to publish, or preparation of the manuscript.

**Competing interests:** The authors have declared that no competing interests exist.

second lockdown can substantially decrease the epidemic peak, the number of multi-infections and deaths.

## Conclusions

Enhanced morbidity and mortality due to multiple disease exposure is a potential threat in the COVID-19 pandemic that deserves more attention. Particularly it underlines another facet questioning disease management strategies aiming for herd immunity.

## Introduction

The Coronavirus Disease 2019 (COVID-19) pandemic caused by the emergence of SARS Coronavirus 2 (SARS-CoV-2) infecting the respiratory system drastically changed the world in 2020. While most cases are asymptomatic or mild, severe infections require hospitalization or even ICU (intensive care unit) treatment, with symptoms of diffuse pneumonia, requiring oxygen supply and mechanical ventilation, potentially causing irreversible health damages or death by multiple organ failure [1].

There is evidence that inadequately protected frontline healthcare workers participated in the rapid initial spread of the pandemic [2], due to inadequate use of personal protective equipment (PPE) or reusing of single-use equipment, which potentially leads to multiple infectious contacts [3]. The mortality among healthcare workers, which is argued to be increased [4], received particular attention [5].

Governments across the globe reacted with different measures to manage the pandemic, many of them being controversial [6] and drastic. While some countries, initially did not strongly interfere with the pandemic, e.g., the UK, [7], or called out for voluntary social distancing, e.g., Sweden, others implemented drastic measures to mitigate the spread of the disease, e.g., Austria, New Zealand. These measures included lockdowns, cancellations of mass gatherings, mandatory social distancing, and isolation interventions that have been successively lifted.

The rationale behind lockdowns is to delay the pandemic to gain the time necessary, to build up healthcare and testing capacities, to develop efficient COVID-19 treatments, and save vaccinations. The latter allows us to immunize the population and avoid a full pandemic peak that would inevitably render the SARS-CoV-2 endemic. This would require constant adaptations of the vaccines as for influenza due to mutation in the virus.

On the contrary, the rationale behind herd immunity is to immunize the population naturally by not intervening with the pandemic [8] and not challenging the economy by drastic interventions. The argument of herd immunity polarizes with advocates in science [8] and politics [9] and also strict opponents [10]. Indeed, the WHO Director-General called the idea of herd immunity unethical [11].

The argument suggesting herd immunity often results from inadequate interpretations of oversimplified simulation models with a lack of in-depth knowledge of healthcare capacities and practices. Thus, overemphasizing the merits of herd immunity while downplaying its dangers. Typically, the implicit assumption of permanent immunity is made, but the decline of antibody levels and cases of COVID-19 reinfections suggest that immunity can be only temporal [12]. Moreover, elevated mortality due to shortages in healthcare resources including hospitals and ICUs is not accounted for and neither is the effect of permanent health damage and their derived long-term medical costs—for example medical costs for illnesses of the

respiratory system in the Federal Republic of Germany amount to almost 5% of the healthcare costs in 2015 (see [13], Chapter 4). Furthermore, indirect costs burdening the social system, e.g., expenses for working disabilities, are nethermost properly addressed. Importantly, an uncontrolled pandemic spread will render the virus endemic, facilitating the emergence and spread of novel variants that differ in their contagiousness and response to antibodies [14–16].

An uncontrolled pandemic spread also facilitates the occurrence of multiple infectious exposures, which might increase the viral load within infections. Indeed, it has been argued that viral load correlates with disease severity, particularly in the context of mandatory usage of facial masks that could prevent from severe infections [17]. Increased viral load might be caused by multiple infectious exposures during the course of a COVID-19 episode [18–21].

A variety of SARS-CoV-2 variants circulate in populations over time [22, 23]. Notably, multiple genetically distinguishable SARS-CoV-2 variants have been detected within the same patient [20, 24, 25]. The global genetic diversity of SARS-CoV-2 is high [25] (although low compared with other RNA viruses [26] due to the proofreading mechanism of the virus used during replication [27]) and the interactions between different viral variants within one organism are insufficiently understood (cf. [27–29]). Most definitely, there is evidence that different viral variants differ in their contagiousness and virulence [30]. It should be mentioned that antibody-dependent enhancement was observed in coronaviruses (including SARS-CoV-2), which leads to more severe episodes due to repeated infectious contacts [31–33].

One of the current challenges in projecting mortality caused by COVID-19 is that estimates are based on data on the initial spread of the pandemic. During this phase, multiple infectious contacts during a disease episode are rare. If mortality is elevated in episodes caused by multiple infectious events with SARS-CoV-2 (multi-infections), predictive models will underestimate this effect if not properly addressed. Remarkably, multiple infectious contacts during one episode are plausible for particular risk groups, such as inadequately protected healthcare workers [34], that are likely to be exposed to multiple infectious contacts within a short period of time and hence become multi-infected.

Here, we introduce a mathematical model to study the effect of increased mortality due to the presence of multiple viral variants due to multiple infectious contacts (multi-infections). The model extends the one from the freely available pandemic preparedness tool CovidSIM [35], which was also generalized in a very different way to study COVID-19 in closed facilities [36] and the effect of antibody-dependent enhancement during vaccination campaigns [37]. As an example the model is parameterized to roughly reflect the situation in the USA. We study the effect of general contact reduction and case isolation measures (which mimic the interventions imposed in the USA) on the extent of multi-infections and its derived increased mortality. In particular, different hypothetical extensions of contact reductions are investigated. Model parameters are chosen from the literature to reflect and adjusted such that the dynamics reflect the situation in the USA until March 2021 (afterwards different hypothetical scenarios are investigated). A range of hypothetical values is chosen for those parameters related to multi-infections, to study the sensitivity of the model to these choices. The presentation follows the structure of [35, 36].

## Methods

We study the hypothetical effect of multiple infections with COVID-19 during one infective period on disease severity and mortality. To do so we adapt the extended SEIR model underlying the pandemic preparedness tool CovidSIM [35]. Our deterministic compartmental model is illustrated in Fig 1. We give a verbal description of the model here and refer to S1 Appendix for a concise mathematical description.

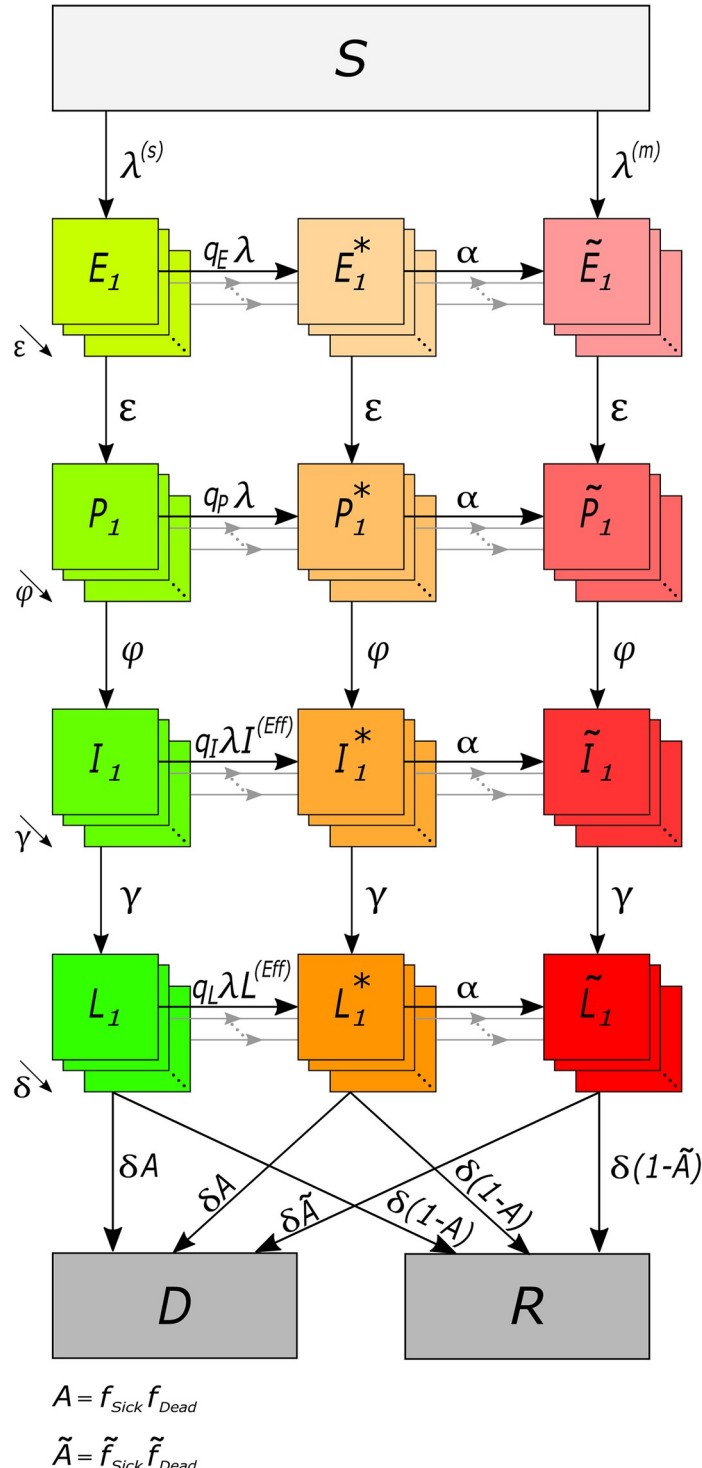

**Fig 1. Model illustration.** Model compartments are shown as boxes. Grey boxes illustrate susceptible ($S$), recovered ($R$), and dead ($D$) individuals. Green boxes illustrate single-infections, orange boxes multi-infections in the transient phase, and red boxes multi-infections characterized by higher morbidity and mortality. Infected compartments consist of latent infected ($E_k$), prodromal ($P_k$), fully contagious ($I_k$), and late infectious ($L_k$) Erlang sub-states. Arrows indicate flows between compartments at the various rates.

A population of *N* individuals is assumed, subdivided into susceptible, infected, and recovered individuals. After a susceptible individual becomes infected it passes through the (i) latency period (the individual is not yet infectious), (ii) prodromal period (the individual does not yet exhibit characteristic symptoms, but is partly infectious), (iii) fully contagious period (the individual is either asymptomatic, or shows symptoms ranging from mild to severe), and (iv) the late infectious period (the individual is no longer fully contagious). After the late infectious phase individuals either recover (and become permanently immune) or they die. The model follows the time change of the numbers of individuals being susceptible (*S*), latently infected (*E*), prodromal (*P*), fully contagious (*I*), and late infectious (*L*), as well as those, who are recovered (*R*) or died (*D*). Deaths unrelated to COVID-19 are ignored, as we assume a pandemic in a large population in a relatively short time period.

A fraction of fully contagious individuals develops symptoms, i.e., they get sick. Asymptomatic infections always lead to recovery, while a fraction of symptomatic infections is lethal. We assume the fractions of symptomatic and lethal infections to be higher among individuals, which got infected multiple times with SARS-CoV-2 during the course of their infection. This reflects multiple viral variants acquired at a single infective event or at consecutive infective events and hence a higher viral load. We therefore follow individuals with 'single' and 'multi' infections throughout all phases of the infection. The numbers of individuals with multi-infections in the various phases are denoted by $\tilde{E}$, $\tilde{P}$, $\tilde{I}$, and $\tilde{L}$. Individuals acquire a multi-infection either at their initial infection, or through consecutive infective events during the course of their infection. The susceptibility to a second infection is different in the latent, prodromal, fully contagious, and late infectious phases. While at infective contacts single-infected individuals spread only single-infection, individuals with multi-infections can spread single- and multi-infections.

Individuals that acquire a multi-infection at their first infective contact can transmit multi-infections, have a higher risk of developing symptoms, and have increased mortality as soon as they reach the respective phase of the disease. However, if individuals move from a single to a multi-infection by a second infective contact, the characteristics of the multi-infection manifest only after a time delay. During this time, individuals are moved into transient compartments $E^*$, $P^*$, $I^*$, and $L^*$. As long as individuals are in the transient phases, they can neither spread multi-infections nor are they more likely to show symptoms or die than individuals with single-infections.

Susceptible individuals acquire infections through contacts with individuals in the prodromal, the fully contagious, or the late infectious periods at rates $\beta_P$, $\beta_I$, $\beta_L$, respectively. Note, a fraction of multi-infected individuals transmits a multi-infection, while the rest transmits only a single-infection.

The basic reproduction number $R_0$ is the average number of infections caused by a single-infected individual in a susceptible population in which no intervention occurs during the whole infectious period, consisting of the prodromal, fully contagious, and late infectious phases. This quantity is assumed to fluctuate seasonally around its annual average $\bar{R}_0$, due to extrinsic factors (weather, seasonal contact behavior, intensity of UV radiation). Importantly, the value of $R_0$ is determined in a virgin population without interventions and thus is determined by normal contact behavior, extrinsic factors, and properties of the virus. In an ongoing epidemic the base reproduction number $R_0$ is replaced by the effective reproduction number, which also accounts for contact-reducing interventions.

The model incorporates general contact reduction measures as well as case isolation. General contact reduction, i.e., curfews or social distancing, reduces the contact rate between all individuals. The intensity of these measures is mainly determined by pandemic management

policies, which change over time. Regarding case isolation, a fraction of symptomatic cases (those that seek medical help) will be isolated in quarantine wards until recovery or death. This fraction is higher in multi- than in single-infections, reflecting more severe symptoms and hence a higher demand for medical help. If the wards are full, individuals are sent into home isolation. While perfect isolation is achieved in quarantine wards, home isolation is imperfect, i.e., not all infectious contacts are avoided.

Classical SEIR models assume that individuals proceed from one infected stage to the next at rates being the reciprocal average residence time spent in each compartment. Implicitly, this assumes exponentially distributed residence times. This is undesirable (cf. [35]), because (i) a proportion of individuals progresses too fast, whereas others progress too slow from one state to the next, and (ii) the variance of the average residence time in each compartment equals its mean. To model more realistic dynamics, we subdivide the latent, prodromal, fully contagious, and late infectious periods into several sub-stages, through which individuals pass successively. As a consequence, residence times follow Erlang-distributions, which are more realistic, because as the means of the residence times does not determine their variances.

### Implementation of the model

The model as described in S1 Appendix was numerically solved by a 4th order Runge-Kutta method. The code was implemented in Python 3.8 using the function solve_ivp as part of the library Scipy, and library Numpy. Graphical output was created using library Matplotlib. The implementation of the model can be found at GitHub (https://github.com/Maths-against-Malaria/MultipleExposuresCOVID19).

### Results

Model parameters are chosen to roughly reflect the situation in the United States of America (USA), where control measures were not as drastic as in other countries and a number of COVID-19 related deaths occurred among frontline healthcare workers [34]. The goal is to study the potential negative effects on morbidity and mortality due to multiple disease exposure during the epidemic peak which is an overlooked risk factor in the debate about herd immunity.

S1–S4 Tables list the parameter values used in the numerical investigations. A population of $N$ = 331 million (reflecting the US) was assumed. The first COVID-19 cases were confirmed in late January 2020 [38], which corresponds to $t$ = 0. For the basic reproduction number we assume an average of $\bar{R}_0 = 3.2$. Seasonality causes it to fluctuate by $a$ = 35% with a peak in late December ($t_{R_{0_{max}}} = 335$). Extrinsic factors such as the intensity of UV radiation, temperature, and time spent inside closed rooms [39] motivate seasonal fluctuations in $R_0$, which are supported by the increase in incidence in seasonal fluctuations in $R_0$ are motivated by fluctuations in the USA and European countries in fall/winter 2020. The average duration of the latent, prodromal, fully contagious and late infectious stages was assumed to be $D_E$ = 3.7, $D_P$ = 1, $D_I$ = 5 and $D_L$ = 5 days, respectively. In the prodromal and late infective stages, individuals were assumed to be half as infective as in the fully contagious stage.

These parameters were justified by CovidSim 1.1 [35] and are a combination of COVID-19 and influenza estimates. (Because the model without multi-infections is essentially equivalent to the one of CovidSim 1.1., the sensitivity of these parameters—and of seasonal fluctuations in $R_0$—can be readily ascertained via the web simulator available at http://version-1.1.covidsim.eu/.)

The general contact reductions for the USA are described in S5 Table. To find parameters properly reflecting general contact-reduction, the age-dependent contact-rate estimates

available from [40] (which averaged over all age strata and weighted by their relative sizes) were used and contact reductions were deduced roughly by the imposed contact restriction in the USA in schools, at work, at home, and other locations. Travel and contact restrictions that reduced contacts by 55% were assumed from day $t_{\mathrm{Dist}_1} = 50$ (early March). These were lifted on day $t_{\mathrm{Dist}_2} = 115$ (mid-May), after which a period of relief (with contact reductions of 22%) was assumed until $t_{\mathrm{Dist}_3} = 190$ (late July). Afterwards, a 'hard lockdown' started, which was as severe as the original restrictions (55%). This lockdown was sustained until October 1st ($t_{\mathrm{Dist}_4} = 255$), followed by a moderate relief (contact reductions of 45%) during the US elections until November 5 ($t_{\mathrm{Dist}_5} = 290$). A second "hard lockdown" (65% contact reductions) was assumed after the US election until thanksgiving holidays ($t_{\mathrm{Dist}_6} = 309$). During the holidays, a relief period (55% contact reduction) was sustained for 7 days (until $t_{\mathrm{Dist}_7} = 316$), after which the hard lockdown was continued with 60% contact reductions until December 10 ($t_{\mathrm{Dist}_8} = 325$). Then a severe pre-holiday-season lockdown started (70% contact reductions) for 10 days (until $t_{\mathrm{Dist}_9} = 335$). Contact reductions were relieved (55%) during the holiday season from December 20 ($t_{\mathrm{Dist}_9} = 335$) until January 8, 2021 ($t_{\mathrm{Dist}_{10}} = 354$). Finally, a hard lockdown with 65% contact reductions was assumed until mid-April 2021 ($t_{\mathrm{Dist}_{11}} = 450$), reflect more restrictive epidemic management under the new administration. For simplicity—and to be able to study the effect of the vaccination—after day $t_{\mathrm{Dist}_{11}} = 450$, all general contact reductions were lifted in the simulations in the default scenario. However, we also assumed alternatives, in which contact reductions were sustained longer.

Multi-infected individuals were more likely to develop severe symptoms (default values: $f_{\mathrm{Sick}} = 58\%$ *vs.* $\tilde{f}_{\mathrm{Sick}} = 64.4\%$) and had an increased mortality (default values: $f_{\mathrm{Dead}} = 4\%$ *vs.* $\tilde{f}_{\mathrm{Dead}} = 5\%$). These parameters, were chosen hypothetically. In the following, we describe the influence of the various model parameters characterizing multi-infection.

The parameter choices lead to dynamics that roughly reflect the number of cases in the USA (cf. Figs 2–7) in the first year of the epidemic. The simulated numbers are higher than those that have been reported in the USA, as they subsume unreported cases.

## Morbidity caused by multi-infections

The increased fraction of developing symptoms in multi-infections compared with single-infections (higher $\tilde{f}_{\mathrm{Sick}}$ in comparison to $f_{\mathrm{Sick}}$) has three effects: (i) slightly fewer people get infected during the pandemic, because of case isolation. In particular, multi-infections cause a higher proportion of symptomatic infections, which are isolated and no longer participate (fully) in transmission. The larger the fraction of multi-infections that become symptomatic, the stronger is this effect. This manifests in higher number of susceptibles (see Fig 2A). (ii) The height of the epidemic peaks is slightly reduced as multi-infections are more likely to be isolated and do not participate in disease transmission. Also, slightly fewer multi-infections occur. This effect manifests during the onset of the epidemic peak around generation $t = 600$, when multi-infections become more frequent (see Fig 2B).

Results without seasonal fluctuations are shown in S1 Fig. Without seasonal fluctuations would have occurred in summer 2020 ($t = 200$) as $R_0$ would haven been much higher leading to a strong increase. These results are shown only to illustrate the effect of seasonal fluctuations. Particularly the contact reductions imposed are meaningless in this case, as contact reductions will be imposed in response to disease incidence, and not independently.

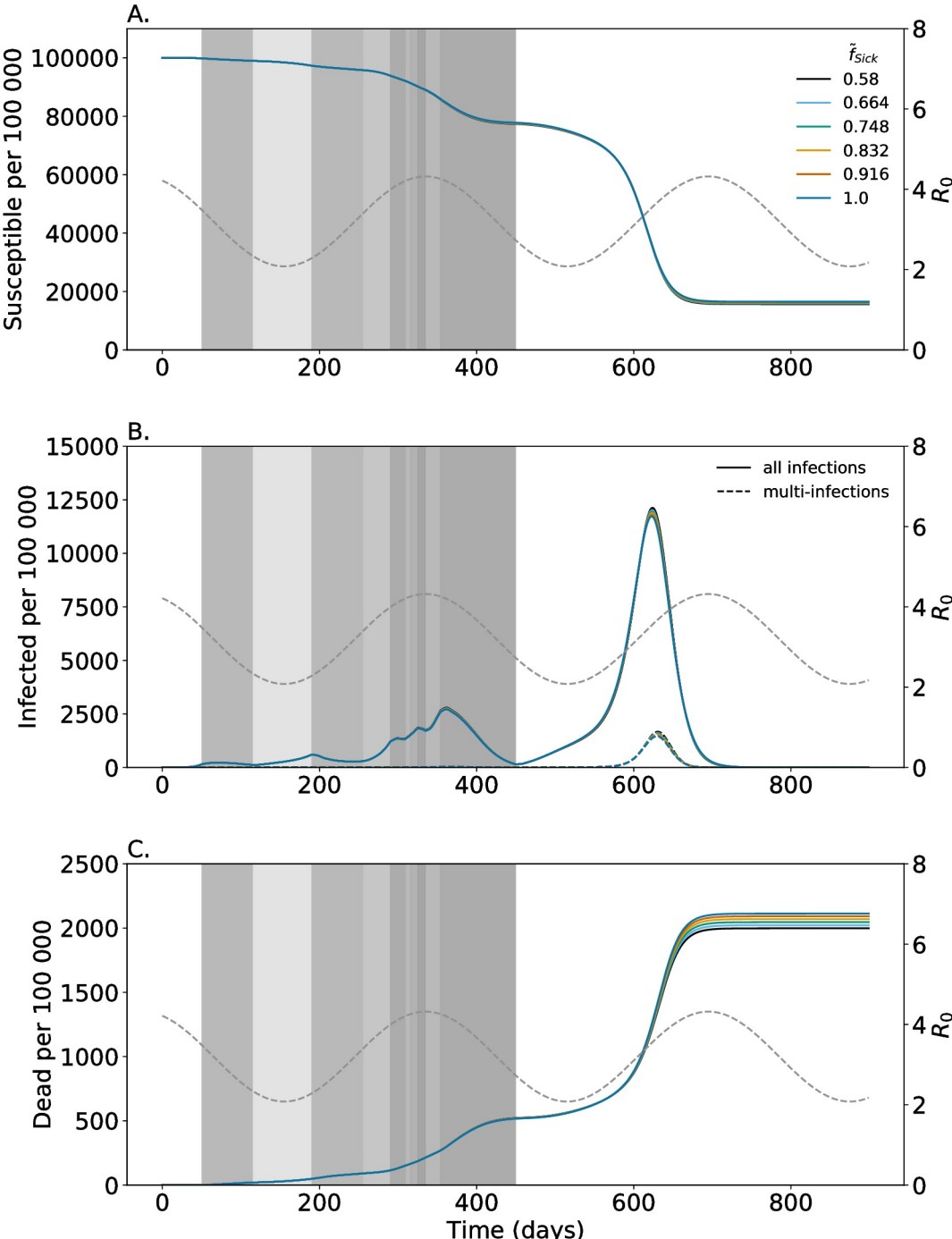

**Fig 2. Proportion of multi-infected individuals being symptomatic.** Shown are the numbers of susceptibles (A); (B) infected individuals consisting of all latent, prodromal, fully, and late contagious individuals among all single- and multi-infections (solid lines) and non-transient multi-infections (dashed lines); and dead individuals (C). Colors are for different fractions of multi-infected individuals that become symptomatic ($\tilde{f}_{\text{Sick}}$). Seasonal fluctuations in $R_0$ are illustrated by the gray dashed line corresponding to the y-axis on the right-hand side. Gray shadings reflect the strength ($p_{\text{Dist}}$) of general distancing in the respective time intervals. Parameters are given in S1–S4 Tables.

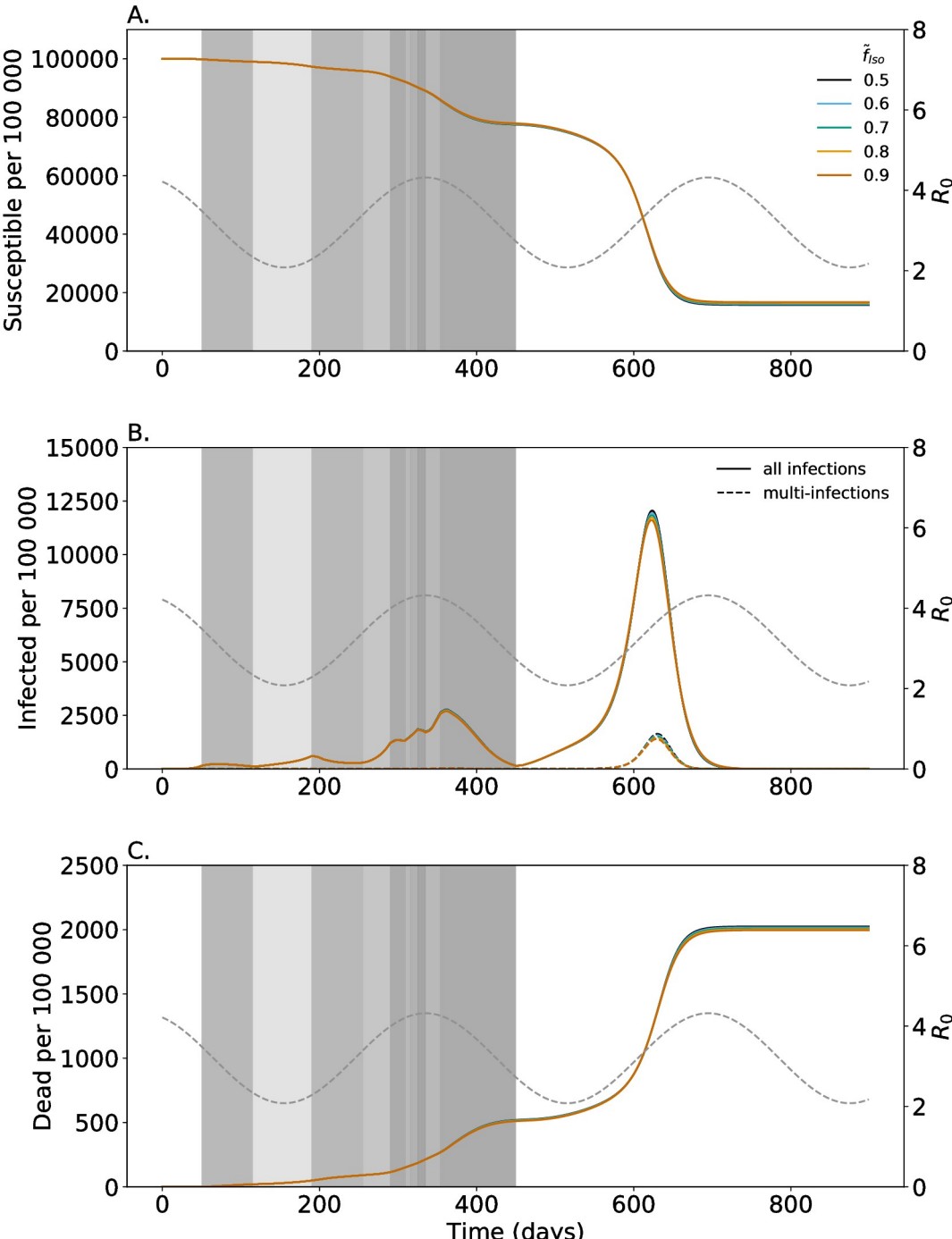

**Fig 3. Proportion of symptomatic multi-infections getting isolated.** As in Fig 2 but for different values $\tilde{f}_{\text{Iso}}$ of multi-infected individuals that become isolated (colors).

## Severity of symptoms

The severity of symptoms is reflected in the model by the fraction of individuals seeking medical help and become isolated. The more severe the symptoms of multi-infections, the more of them get isolated (higher $\tilde{f}_{\text{Iso}}$). Isolating a higher fraction of symptomatic multi-infections

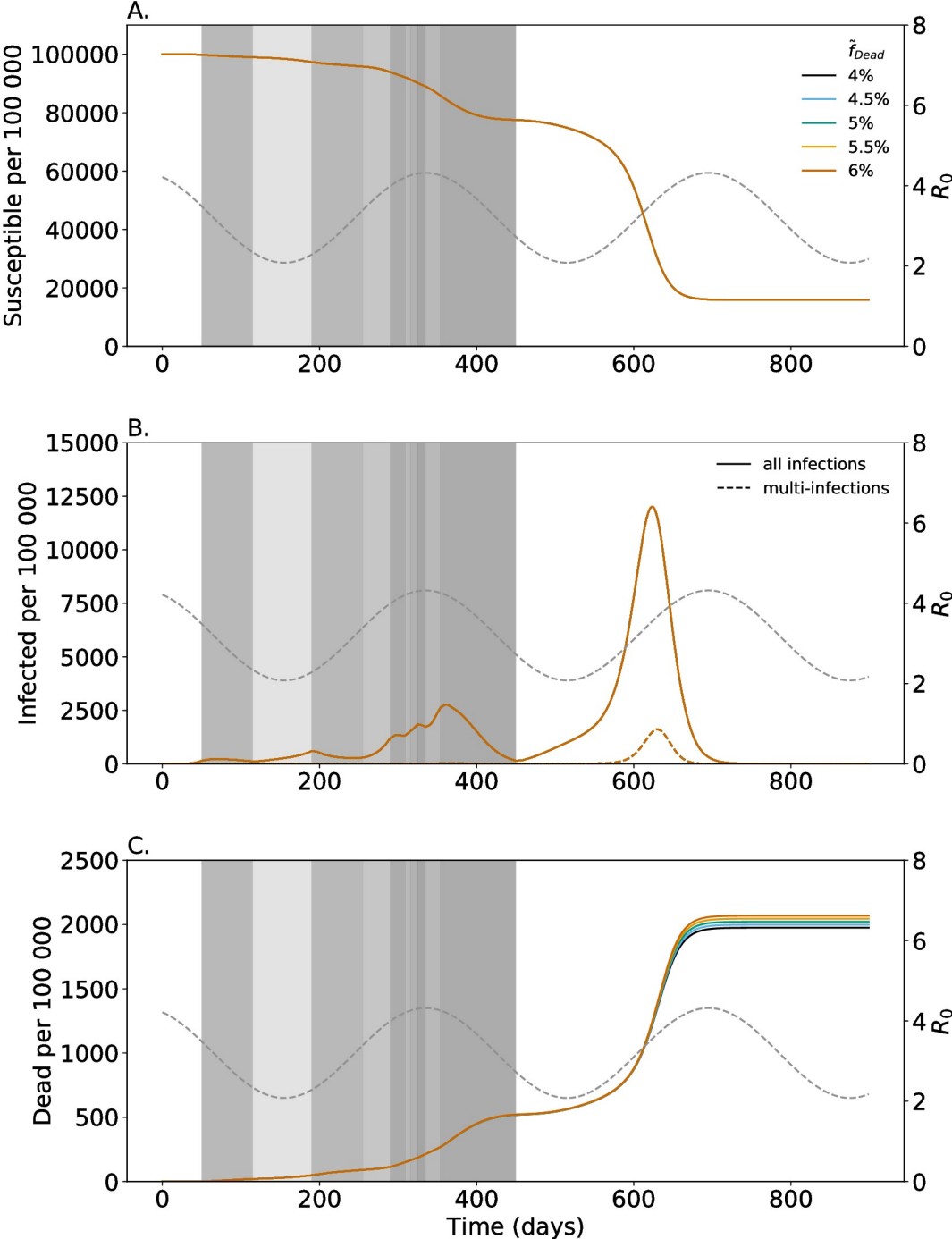

**Fig 4. Increased mortality of symptomatic multi-infections.** As in Fig 2 but for different values of increased mortality ($\tilde{f}_{Dead}$) of symptomatic multi-infected individuals (colors).

leads to slightly fewer infections and a reduced height of the epidemic peaks (Fig 3A). Also, the peak number of multi-infections decreases (Fig 3B). Moreover, there is a reduction in mortality (Fig 3C). However, this reduction does not compensate the potential increase in mortality due to multi-infections (compare Fig 2C with Fig 3C).

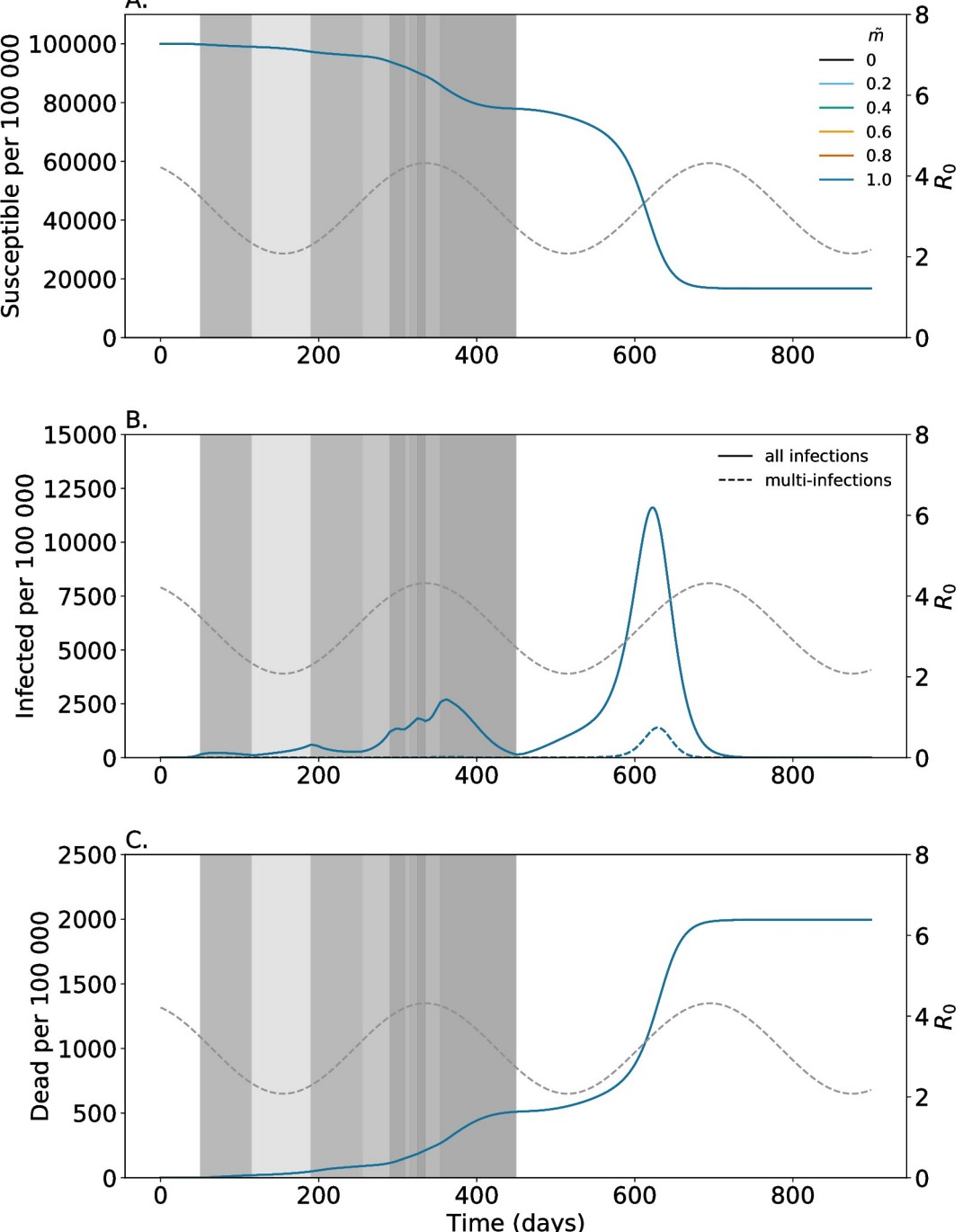

**Fig 5. Likelihood to spread multi-infections.** As in Fig 2 but colors are for different likelihoods of multi-infected individuals to spread multi-infections ($\tilde{m}$).

## Increased mortality

As described above increased morbidity of multi-infections reflected by lager fractions of symptomatic and isolated infections decreases the overall number of cases, but causes a higher

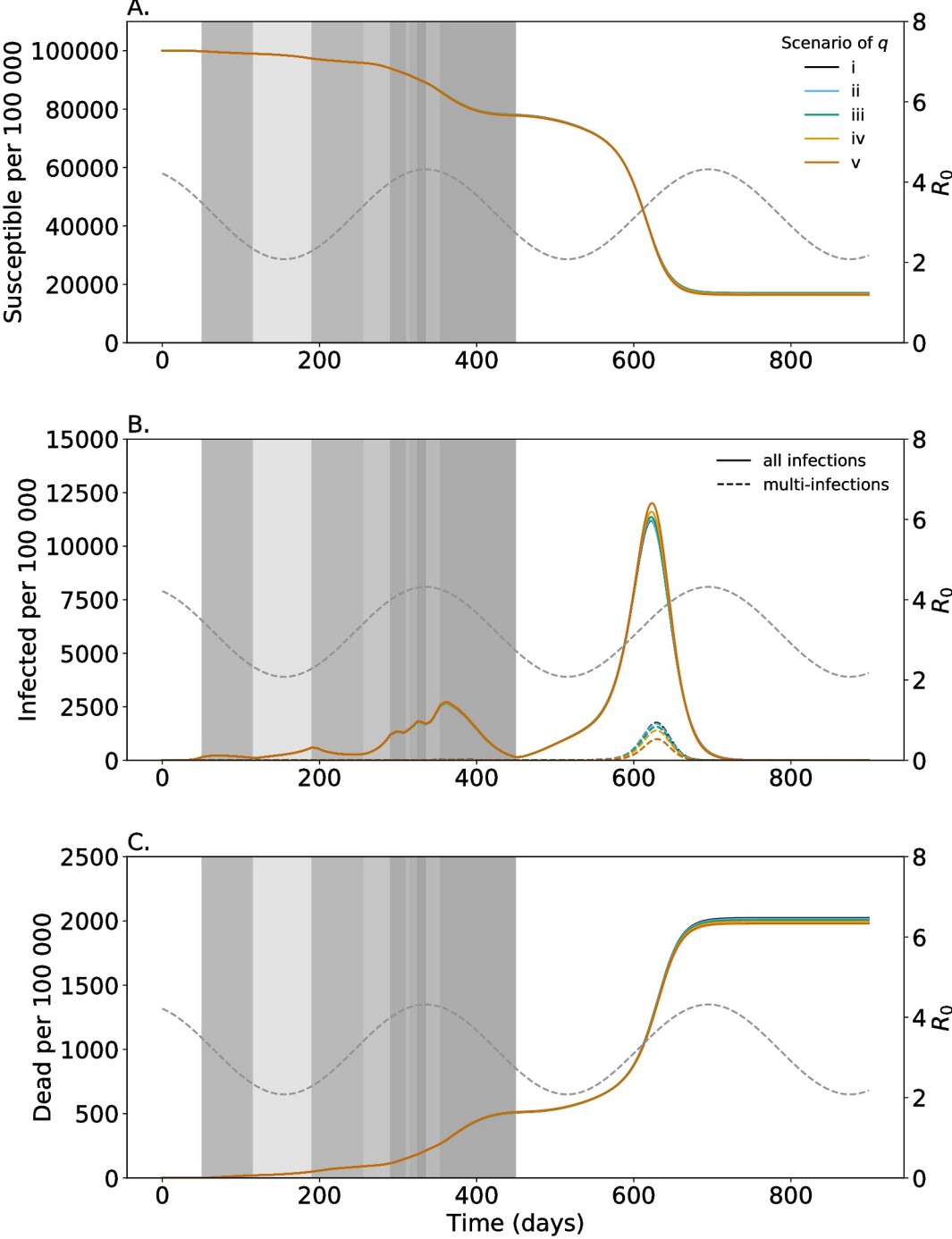

**Fig 6. Susceptibility to successive infections.** As in Fig 2 but colors are for the different scenarios (i-v) of susceptibility to multi-infections listed in Table 1.

number of deaths. This increase in mortality scales with the case fatality of multi-infections reflected by the parameter $\tilde{f}_{\text{Dead}}$. As shown in Fig 4 this parameter does not affect the number of infections, but the number of lethal infections. The higher case mortality of multi-infections, the higher the number of deaths.

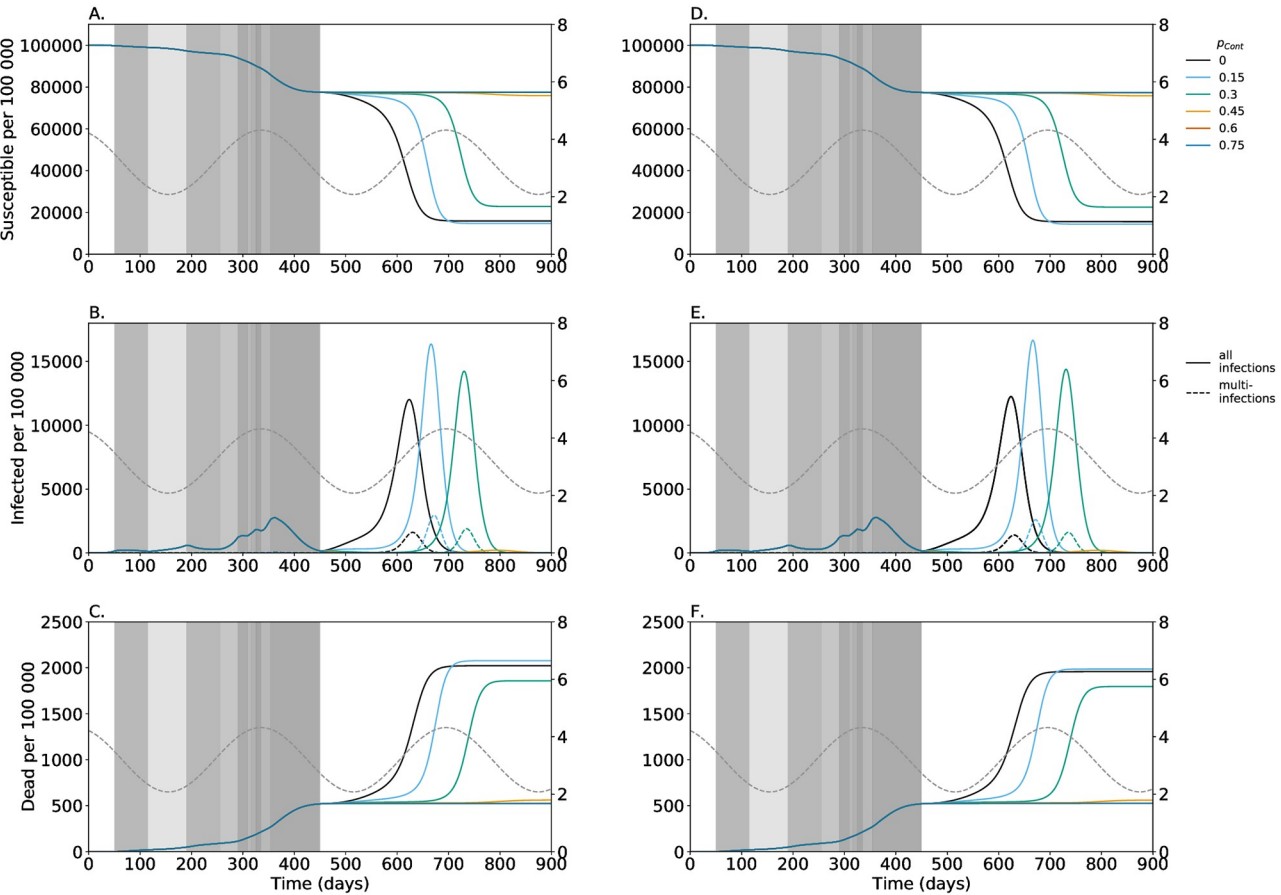

**Fig 7. Implication of general distancing after day 450—Considering multiple-infections having an effect or not.** As in Fig 2 but colors are for different levels of general contact reduction after day $t$ = 450. Panels A-C assume assume a higher (i) likelihood of symptomatic infections ($\tilde{f}_{\text{Sick}} > f_{\text{Sick}}$), (ii) higher mortality ($\tilde{f}_{\text{Dead}} > f_{\text{Dead}}$), (iii) more pronounced symptoms $\tilde{f}_{\text{Iso}} \neq f_{\text{Iso}}$, and a higher likelihood of causing multi-infections ($\tilde{m} = 0.2$), whereas panels D-F neglect these effects ($\tilde{f}_{\text{Sick}} = f_{\text{Sick}}, \tilde{f}_{\text{Dead}} = f_{\text{Dead}}, \tilde{f}_{\text{Iso}} = f_{\text{Iso}}$, and $\tilde{m} = 0$).

## Likelihood of spreading multi-infections

Multi-infections are acquired either at a single infective event or at consecutive infective events. In case of the latter, the negative effects in terms of morbidity and mortality only manifest after a transient phase. In particular, in this phase individuals cannot spread multi-infections and they might recover before the negative effects of multi-infections are in place. The likelihood of multi-infected individuals to spread multi-infections ($\tilde{m}$) only marginally effects the number of infections during the epidemic (Fig 5A and 5B). Also, the effect on mortality is small.

## Susceptibility to successive infections

Individuals already infected with COVID-19 will show an immune reaction that mediates susceptibility to successive infections. Thus, the force of infection experienced by single-infected individuals might be reduced compared with that acting on susceptible. Namely, only percentages $q_E, q_P, q_I, q_L$ of the force of infection are acting on single-infected individuals in the latent, prodromal, fully contagious, and late infectious stages, respectively. Typically, susceptibility to further infections should not be affected much in the latent phase, while it is strongly decreased

**Table 1. Susceptibility to successive infections.**

| Scenario | $q_E$ | $q_P$ | $q_I$ | $q_L$ |
|----------|-------|-------|-------|-------|
| i | 1 | 1 | 1 | 1 |
| ii | 1 | 1 | 1 | 0.75 |
| iii | 1 | 1 | 0.75 | 0.5 |
| iv | 1 | 0.75 | 0.5 | 0.25 |
| v | 0.75 | 0.5 | 0.25 | 0 |

Different scenarios for susceptibility to successive infections in different stages of the infection.

in the late infectious phase, in which some immunity is already developed. Clearly susceptibility to successive infections affects the number of multi-infections. We studied the five scenarios listed in Table 1. They correspond to decreasing susceptibility, with scenario (i) being the extreme that susceptibility is not affected, and scenario (iv) being the default used in the simulations here.

The number of infections slightly increases for decreasing susceptibility (Fig 6A). Also, the epidemic peak is higher, while the number of multi-infections changes dramatically (Fig 6B). The reason is as follows: the number of multi-infections is highest shortly after the epidemic peak when most infections are single-infection, and it is thus more likely to get infected twice during the course of the disease than once by a multi-infected individual. Reduced susceptibility of single-infected individuals results in a clear reduction of multi-infections. Which then result in a reduced mortality (Fig 6C).

## The effect of countermeasures

For the first year of the pandemic general distancing follow roughly the interventions imposed by the US government. Until day $t = 450$ we anticipated further contact reduction, following a subsequent adjustment. For all scenarios described above, no general distancing after day 450 was assumed. This allows to anticipate how the infections, especially multi-infections, would develop without strong interventions.

To guide future policies, it was also tested, which intensity of general distancing would be needed beyond day $t = 450$ to keep incidence low. Fig 7A–7C shows the outcome of simulations assuming five different levels of general contact reduction beyond day $t = 450$ until the end of the simulations. The effects fall into two categories: (i) for $p_{\mathrm{Dist}}$ below 30% a third wave occurs in the next winter; (ii) for $p_{\mathrm{Dist}}$ above 45% incidences stay low throughout the simulated interval until day 900. The effect of general distancing results in a delay of the epidemic peak, whose height depends on seasonal fluctuations in $R_0$, with higher $R_0$ leading to a higher peak (cf. [35]). The less restrictive distancing after $t = 450$, the earlier the "third wave". Without any distancing it would peak before the seasonal effect of $R_0$ is maximal, while low levels of distancing would delay the peak until the onset of the flue season. This would result in higher numbers of infections and higher mortality. In this case multi-infections are disproportionally increased as multi-infections mostly occur if many individuals are infected at the same time. Also, a contact reduction of $p_{\mathrm{Dist}} = 30\%$ after day $t = 450$, yield a higher epidemic peak compared with the default case of no contact reductions after $t = 450$. However, the peak occurs in spring 2022 when $R_0$ is declining, so that the overall number of deaths is lower. Nevertheless, high epidemic peaks are undesirable, because they overwhelm the health care system. For distancing of $p_{\mathrm{Dist}} = 45\%$ the number of infections are low almost until the end of the simulation period but a peak may occur later.

Also, the compound effects of multi-infections were investigated by contrasting the scenarios of Fig 7A–7C with the analogous situations in which multi-infections do not differ from single-infections in terms of morbidity ($\tilde{f}_{\text{Sick}} = f_{\text{Sick}}$), mortality ($\tilde{f}_{\text{Dead}} = f_{\text{Dead}}$), symptoms resulting in isolation ($\tilde{f}_{\text{Iso}} = f_{\text{Iso}}$), and in causing multiple-infections ($\tilde{m} = 0$), as shown in Fig 7D–7F. The compound effects of multi-infections lead to slightly lower epidemic peaks, slightly fewer infections, but considerably higher mortality (cf. Fig 7A–7C with Fig 7D–7F). The differences are illustrated in more detail in S2 Fig.

## Discussion

A substantial number of COVID-19 infections was reported among frontline healthcare workers [5, 34, 41]. Particularly, their role in initially spreading the pandemic was recognized [42]. Partly, this is due to improper use of personal protective equipment (PPE) [3, 43]. Frontline healthcare workers are exposed (i) relatively long to infected individuals, and (ii) to multiple infective contacts with different patients within a short time period. Such infections can lead to a higher viral load upon infection [18], or super-infections with different viral variants during the course of the disease [20]. Notably, higher viral loads have been associated with more severe disease [17, 44]—in line with a high number of severe infections among healthcare workers [45]. A similar logic applies to persons in initial epicenters of the global pandemic, e.g., in skiing resorts in northern Italy [46, 47]. Multiple infectious contacts and longer (average) exposure to the virus is initially restricted to certain risk groups, but will become common during the pandemic peak. This is an overseen threat by advocates of herd immunity.

Viral diversity is so high that individuals acquire different viral variants at different infective events. Another overseen risk factor is the increasing viral diversity during an epidemic (cf. [30, 48])—leading to frequent infections with different viral variants when the pandemic peaks.

We studied the potential effect of increased morbidity and mortality due to 'multi-infections', i.e., infections characterized by a higher viral diversity or higher viral load—acquired either upon infection or during successive infections during one disease episode. For this purpose, we adapted the model underlying the pandemic preparedness tool CovidSim (cf. [35], http://covidsim.eu/), which incorporates general contact reduction during 'lockdowns' and case isolation. The model was extended to distinguish between single-infections, i.e., infections characterized by less viral diversity and lower viral load upon infection, and multi-infections. The latter are characterized by higher morbidity (i.e., more infections are symptomatic, and more symptomatic infections need medical care) and mortality (i.e., more symptomatic infections are lethal).

Although the disease is age-dependent this factor can be overlooked when considering: (i) an industrial country, in which the herd immunity strategy was particularly controversial, since these countries have the financial means to implement strong and efficient response strategies to COVID-19, (ii) the average proportion of being symptomatic or having severe infections. In populations, in which age structure can be neglected this is sufficiently accurate. For models with an explicit age structure we refer to [49, 50]. In general, age structure can be modeled explicitly in a rather simple way. Important is that the contact behavior between age groups is modeled explicitly (not assumed to be random—such a model would be equivalent to a model without age dependency, but stratified into age groups), (iii) the complexity of the model, as $R_0$ has to be defined by the next-generation matrix (which can be done rather easily), but the description becomes much more complex. Our aim was to show the potential dangers of herd-immunity strategies for which age-dependency is not necessarily required.

We numerically investigated potential model dynamics reflecting the situation in the USA under a range of parameters summarizing morbidity and mortality of multi-infections.

Higher morbidity associated with multi-infections leads to a reduction of episodes (if case isolation is sustained)—multi-infections are more severe and thus more likely be detected and isolated. However, the reduction in the number of infections has to be interpreted with caution: overall mortality is still higher due to multi-infections. Moreover, more infections are severe leading to an increased demand for medical care and potentially long-term health damages. Estimates of case fatality and disease severity at the onset of a pandemic, when multi-infections are rare, will therefore be underestimated. This also includes the estimated demand for medical capacities, treatment costs, and long-term costs due to permanent health damages. Our results show that depending on the level of further general distancing, epidemic peaks may be shifted into the flu season, leading to higher (and narrower) peaks. This disproportionally increases the number of multi-infections and hence deaths. The effects are mediated by (i) the increase in mortality in multi-infections, (ii) the likelihood of acquiring a second infection during the various infected phases (before immunity) is reached, and (iii) the increase in the likelihood of developing symptoms. Concluding, increased mortality due to multi-infections is a potential threat.

We adjusted model parameters such that they yielded roughly the epidemic dynamics in the USA in the first year of the epidemic. After spring 2021, we investigated different levels of general contact reductions. Importantly, as default, we assumed no contact reductions after spring 2021 to illustrate how the epidemic would develop without interventions (reflecting a strategy aiming for herd immunity, i.e., no general contact reductions but sustained case isolation). Comparing this scenario with different levels of sustained contact reductions illustrated the importance of seasonal fluctuations in the base reproductive number. Namely, general contact reductions, shift the epidemic peaks. An epidemic peak that is shifted into the onset of the flu season is undesirable, as it will lead to a higher and narrower peak. In such a situation, multi-infections might lead to increased overall mortality, besides the fact that the healthcare system is much more challenged. Shifting epidemic peaks into the fading flu season leads to a substantial reduction of the number of cases, multi-infections, and mortality. The simulations have to be interpreted with caution. Seasonal fluctuations are unpredictable as well as human behavior during a pandemic. Therefore, our results elucidate qualitative mechanism rather than precise quantitative predictions.

Our results are intended to demonstrate the hypothetical effect of multi-infections in simple and straightforward setups. We assume that individuals become permanently immune if they do not die of COVID-19. This neglects temporal immunity [31, 51]. Therefore, the number of superinfections and derived deaths can further increase. Moreover, we did not consider self-imposed distancing measures, i.e., as the number of cases increases, individuals will avoid contacts, have a higher propensity to wear facial masks, etc. This has two reasons: (i) anticipation of human behavior would be speculative and can never be adequately captured by a model; (ii) we wanted to explore the effect in a scenario that aims for herd immunity. Hence, we only wanted to incorporate the most evident control interventions. Nevertheless, it is easy to adapt the model to incorporate more control interventions.

Empirical evidence for multi-infections is notoriously difficult to obtain at the beginning of a pandemic. Although the number of COVID-19 infections is threatening, at an early stage, without a complete understanding of the pathogenesis, it is impossible to design a proper study of multi-infections. Namely, a representative sample size cannot be achieved despite the many confounding factors.

Notably, multi-infections are not the only danger when aiming for herd immunity. An uncontrolled (or hardly controlled) pandemic inevitably renders the virus endemic. This will

challenge future SARS-CoV-2 eradication campaigns, as also animals (including, e.g., cats and dogs) become a reservoir for transmission that is difficult to control. Moreover, viral diversity will increase, and aggressive forms of the virus might spread. Viral diversity also challenges vaccination campaigns. Namely, vaccines will need to be adjusted yearly, as it is the case for flu vaccines. Thus, many rounds of vaccinations will be necessary to eradicate the virus. This is particularly dangerous in the context of antibody-dependent enhancement, which has been reported in corona viruses, and is thought to be a potential obstacle in vaccine development [52, 53].

Summarizing, increased morbidity and mortality due to multi-infections is an important but overseen risk, particularly in the context of herd immunity. Multi-infections will become more relevant during the course of the pandemic, when viral diversity and disease prevalence increases. The UK variant which occur as a result of viral mutation in an infected individual spreads faster and causes higher morbidity and mortality. This strain later spreads to several other countries such as Australia and some European countries (Denmark, Spain, France etc.). The variant has been reported to be from a traveler from the UK. This strain is a threat to achieving herd immunity because of the rate at which it spreads and the morbidity and mortality it causes. Altogether, evidence-based research on multi-infections is necessary.

## Supporting information

**S1 Fig. Morbidity caused by multi-infections without seasonal fluctuations.** The same as Fig 2 but without seasonal fluctuations.
(PDF)

**S2 Fig. Characteristics of multi-infections.** Shown are the total numbers of (A) deaths; (B) case fatality (i.e., deaths per infected individuals); (C) total percentage (of the population) being infected or multi-infected; and (D) the maximal number of individuals infected or multi-infected at certain census points. For each value of general distancing $p_{\text{Dist}}$ after day $t$ = 450 values are shown under the assumptions that: (a) multi-infections cause higher morbidity and mortality, and are more likely to be isolated than single-infections (corresponding to Fig 7A–7C); (b) morbidity, mortality, and isolation are the same as for single-infections (corresponding to Fig 7D–7F). The horizontal lines in scenario (a) in (C) and (D) indicate the number of multi infections characterized by increased morbidity and mortality given by (33) in S1 Appendix. Parameters are given in S4 and S5 Tables.
(PDF)

**S1 Appendix. Mathematical description.**
(PDF)

**S1 Table. Population size and model compartments.**
(PDF)

**S2 Table. Summary of model parameters describing disease progression and choices for the simulations.**
(PDF)

**S3 Table. Summary of model parameters describing infections and choices for the simulations.**
(PDF)

**S4 Table. Summary of model parameters and choices for the simulations.**
(PDF)

**S5 Table. Contact reduction parameters (reflecting roughly the policies in the USA).** (PDF)

## Acknowledgments

We want to dedicate this work to all the victims of the SARS-CoV-2 virus. Our grief is with friends and families of the dreadful disease. The authors like to express their sympathy to all working to find a cure for the virus. The authors gratefully acknowledge the helpful comments and discussions with Prof. Martin Eichner on the model. The authors are thankful for the constructive comments of an anonymous reviewer.

## Author Contributions

**Conceptualization:** Arlinda Sadiku, Girma Mesfin Zelleke, Toheeb Babatunde Ibrahim, Aliou Bouba, Vincent Appiah, Gideon Akumah Ngwa, Miranda Ijang Teboh-Ewungkem, Kristan Alexander Schneider.

**Formal analysis:** Kristina Barbara Helle, Arlinda Sadiku, Girma Mesfin Zelleke, Toheeb Babatunde Ibrahim, Aliou Bouba, Kristan Alexander Schneider.

**Funding acquisition:** Kristan Alexander Schneider.

**Investigation:** Kristina Barbara Helle, Arlinda Sadiku, Toheeb Babatunde Ibrahim, Aliou Bouba, Vincent Appiah, Kristan Alexander Schneider.

**Methodology:** Kristina Barbara Helle, Arlinda Sadiku, Girma Mesfin Zelleke, Toheeb Babatunde Ibrahim, Aliou Bouba, Vincent Appiah, Gideon Akumah Ngwa, Miranda Ijang Teboh-Ewungkem, Kristan Alexander Schneider.

**Project administration:** Aliou Bouba, Kristan Alexander Schneider.

**Resources:** Kristan Alexander Schneider.

**Software:** Kristina Barbara Helle, Girma Mesfin Zelleke, Toheeb Babatunde Ibrahim, Aliou Bouba, Henri Christian Tsoungui Obama, Junior, Kristan Alexander Schneider.

**Supervision:** Kristan Alexander Schneider.

**Validation:** Kristan Alexander Schneider.

**Visualization:** Kristina Barbara Helle, Arlinda Sadiku, Toheeb Babatunde Ibrahim, Aliou Bouba, Kristan Alexander Schneider.

**Writing – original draft:** Kristina Barbara Helle, Arlinda Sadiku, Girma Mesfin Zelleke, Toheeb Babatunde Ibrahim, Aliou Bouba, Vincent Appiah, Miranda Ijang Teboh-Ewungkem, Kristan Alexander Schneider.

**Writing – review & editing:** Kristina Barbara Helle, Arlinda Sadiku, Girma Mesfin Zelleke, Toheeb Babatunde Ibrahim, Aliou Bouba, Henri Christian Tsoungui Obama, Junior, Vincent Appiah, Gideon Akumah Ngwa, Miranda Ijang Teboh-Ewungkem, Kristan Alexander Schneider.

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
