## [Decision Letter · Decision Letter 0]

8 Feb 2021

PONE-D-20-32872

Is increased mortality by multiple exposures to COVID-19 an overseen factor when aiming for herd immunity?

PLOS ONE

Dear Dr. Schneider,

Thank you for submitting your manuscript to PLOS ONE. After careful consideration, we feel that it has merit but does not fully meet PLOS ONE’s publication criteria as it currently stands. Therefore, we invite you to submit a revised version of the manuscript that addresses the points raised during the review process.

Your manuscript was reviewed by one expert in the field. Many potential reviewers did not accept the review invitation. The reviewer identified several important problems in your submission which require your careful attention before the final decision can be made. It is important that you carefully consider all comments and provide detailed point-by-point responses.

We look forward to receiving your revised manuscript.

Kind regards,

Yury E Khudyakov, PhD

Academic Editor

PLOS ONE

Journal Requirements:

2.In your Data Availability statement, you have not specified where the minimal data set underlying the results described in your manuscript can be found. PLOS defines a study's minimal data set as the underlying data used to reach the conclusions drawn in the manuscript and any additional data required to replicate the reported study findings in their entirety. All PLOS journals require that the minimal data set be made fully available. For more information about our data policy, please see http://journals.plos.org/plosone/s/data-availability.

3.We suggest you thoroughly copyedit your manuscript for language usage, spelling, and grammar. If you do not know anyone who can help you do this, you may wish to consider employing a professional scientific editing service.  

4. Please ensure that the abstract and introduction are written in an objective manner, and that statements are adequately referenced.

Reviewers' comments:

Reviewer's Responses to Questions

**Comments to the Author**

1. Is the manuscript technically sound, and do the data support the conclusions?

Reviewer #1: Yes

2. Has the statistical analysis been performed appropriately and rigorously? 

Reviewer #1: Yes

3. Have the authors made all data underlying the findings in their manuscript fully available?

Reviewer #1: No

4. Is the manuscript presented in an intelligible fashion and written in standard English?

Reviewer #1: Yes

5. Review Comments to the Author

Reviewer #1: In the paper, the authors establish a model for Covid-19 multi-infections using the published CovidSim tool and use their model to observe how multi infections will affect the population's morbidity and mortality rates. The authors use the model for different scenarios of seasonal changes and different forms of lock downs. The population is split into susceptible, infected, and recovered individuals in a proposed deterministic compartment model. Further, infected individuals are partitioned into different infection stages with different infection and transition rates. The author has discussed three types of infections: single infection, second is a transient infection, and the last one is multi-infection. Transient infections are infections in which individuals are exposed to infected individuals multiple times and acquire single infections numerous times within a single infection period. Multi-infection implies infection with multiple viral variants or multiple viral loads in comparison to a single infection.

Overall, the study is interesting and the model adds to previously published models. However, some of the following points need to be addresses:

1. There are many constants relating to the disease spread that are assumed throughout the study and the authors should provide a rationale for most of them or some reference to why these were chosen.

2. Some of the legends are missing in the figures (such as the dashed lines) and these should be added.

3. The disease caused by the virus is very age dependent. However, the authors did not give any consideration to different probabilities due to different ages. This should at least be discussed in the discussion section. See for example models Mizrahi, Stern, Open Biology. Balabdaoui, Mohr Scientific Reports.

4. Table 2 is hard to understand what each column represents.

5. Ro is relatively high for todays’ measures of social distancing and face masks. The authors address this in the discussion but it would be good to perhaps add in the supplementary a scenario where social distancing and masks are kept.

6. The reduction in the interactions in lock down scenarios is relatively mild (40%-60%). What happens in a more tight lock down?

7. How likely is a multi-infection scenario given that most people stay within their area in these times and at the same area usually there is the same strain of the virus? Maybe it would be good to add references about local different strains.

6. PLOS authors have the option to publish the peer review history of their article (what does this mean?). If published, this will include your full peer review and any attached files.

Reviewer #1: No

---

## [Decision Letter · Decision Letter 1]

20 May 2021

PONE-D-20-32872R1

Is increased mortality by multiple exposures to COVID-19 an overseen factor when aiming for herd immunity?

PLOS ONE

Dear Dr. Schneider,

Thank you for submitting your manuscript to PLOS ONE. After careful consideration, we feel that it has merit but does not fully meet PLOS ONE’s publication criteria as it currently stands. Therefore, we invite you to submit a revised version of the manuscript that addresses the points raised during the review process.

I would like you to verify accuracy and relevance of all your references.

Some references like 19 and 20 seem to be irrelevant to the points stated in the text.

Ref 24 is published in PloS One and does not describe ADE but uses it in the model.

Reference are required for the following statements: “Increased viral load might be caused by multiple infectious exposures  during the course of a COVID-19 episode”, and “Such infections can lead to a higher viral load upon infection or super-infections with different viral variants during the course of the disease.”

Some references like 16 and 17 are outdated. More relevant papers have been recently published about genetic diversity.  

Some references like 31 are incomplete

We look forward to receiving your revised manuscript.

Kind regards,

Yury E Khudyakov, PhD

Academic Editor

PLOS ONE

Journal Requirements:

Reviewers' comments:

Reviewer's Responses to Questions

**Comments to the Author**

1. If the authors have adequately addressed your comments raised in a previous round of review and you feel that this manuscript is now acceptable for publication, you may indicate that here to bypass the “Comments to the Author” section, enter your conflict of interest statement in the “Confidential to Editor” section, and submit your "Accept" recommendation.

Reviewer #1: All comments have been addressed

2. Is the manuscript technically sound, and do the data support the conclusions?

Reviewer #1: Yes

3. Has the statistical analysis been performed appropriately and rigorously? 

Reviewer #1: Yes

4. Have the authors made all data underlying the findings in their manuscript fully available?

Reviewer #1: Yes

5. Is the manuscript presented in an intelligible fashion and written in standard English?

Reviewer #1: Yes

6. Review Comments to the Author

Reviewer #1: (No Response)

7. PLOS authors have the option to publish the peer review history of their article (what does this mean?). If published, this will include your full peer review and any attached files.

Reviewer #1: No

---

## [Editor Report · Decision Letter 2]

14 Jun 2021

Is increased mortality by multiple exposures to COVID-19 an overseen factor when aiming for herd immunity?

PONE-D-20-32872R2

Dear Dr. Schneider,

We’re pleased to inform you that your manuscript has been judged scientifically suitable for publication and will be formally accepted for publication once it meets all outstanding technical requirements.

Kind regards,

Yury E Khudyakov, PhD

Academic Editor

PLOS ONE
---

## [Editor Report · Acceptance letter]

30 Jun 2021

PONE-D-20-32872R2 

Is increased mortality by multiple exposures to COVID-19 an overseen factor when aiming for herd immunity? 

Dear Dr. Schneider:

I'm pleased to inform you that your manuscript has been deemed suitable for publication in PLOS ONE. Congratulations! Your manuscript is now with our production department. 

Kind regards, 

on behalf of

Dr. Yury E Khudyakov 

Academic Editor

PLOS ONE